

# Mechanisms of methane transport through *Populus trichocarpa*

Ellynne Kutschera[1], Aslam Khalil[1], Andrew Rice[1], and Todd Rosenstiel[2]

[1]Portland State University, Department of Physics, P.O. Box 751, Portland, OR 97201
[2]Portland State University, Department of Biology, P.O. Box 751, Portland, OR 97201

*Correspondence to:* Ellynne Kutschera (ekutsche@pdx.edu)

**Abstract.** Although the dynamics of methane ($CH_4$) emission from croplands and wetlands have been fairly well investigated, the contribution of trees to global $CH_4$ emission and the mechanisms of tree transport are relatively unknown. $CH_4$ emissions from the common wetland tree species *Populus trichocarpa* (black cottonwood) native to the Pacific Northwest were measured under hydroponic conditions in order to separate plant transport mechanisms from the influence of soil processes. Roots were exposed to $CH_4$ enriched water and canopy emissions of $CH_4$ were measured. The average flux for 34 trials (at temperatures ranging from 17 to 25 $^o$C) was $2.8 \pm 2.2$ $\mu$g $CH_4$ min$^{-1}$ (whole canopy). Flux increased with temperature. Compared to the isotopic composition of root water $CH_4$, $\delta^{13}$C values were depleted for canopy $CH_4$ where the warmest temperatures (24.4-28.7 $^o$C) resulted in an epsilon of $2.8 \pm 4.7‰$; midrange temperatures (20.4- 22.1 $^o$C) produced an epsilon of $7.5 \pm 3.1$ ‰; and the coolest temperatures (16.0-19.1 $^o$C) produced an epsilon of $10.2 \pm 3.2$ ‰. From these results it is concluded that there are likely multiple transport processes at work in $CH_4$ transport through trees and the dominance of these processes changes with temperature. The transport mechanisms that dominate at low temperature and low flux result in a larger fractionation, while the transport mechanisms that prevail at high temperature and high flux produce a small fractionation. Further work would investigate what combination of mechanisms are specifically engaged in transport for a given fractionation of emitted $CH_4$.

## 1 Introduction

A significant portion of the global cycle of methane ($CH_4$) takes place in terrestrial ecosystems, although it is its importance as a greenhouse gas that garners attention. $CH_4$ is second only to $CO_2$ in terms of the radiative forcing of greenhouse gasses at current concentrations (Myhre et al., 2013). Although the rate of increase of $CH_4$ in the atmosphere has slowed, atmospheric $CH_4$ concentrations continue to rise (Ciais et al., 2013). Uncertainties in emissions estimates persist, but the largest single source of $CH_4$ emissions is natural wetlands (Ciais et al., 2013). In one study of a fen ecosystem, up to 90% of $CH_4$ emission was observed to be through plants (Whiting and Chanton, 1992). $CH_4$ emitted through trees in flooded systems may contribute a significant fraction of these wetland emissions (Rice et al., 2010; Pangala et al., 2013). Whether in wetlands or in upland soils, it is not yet known if transport of $CH_4$ through trees is significant in the overall movement of $CH_4$ through an ecosystem. The transport mechanisms of $CH_4$ through trees is potentially a controlling factor for the overall $CH_4$ flux and should be understood.



In his review of isotopic signatures for $CH_4$ transport through wetlands, Chanton (2005) identifies three different pathways: transport through the water layers separating soil and air, ebullition or bubbling from submerged soils across the water layer to the air, and transport through emergent aquatic plants. Of the types of transport mechanisms described in the literature for plants and trees, transpiration and aerenchyma formation have been significantly discussed for trees. Aerenchyma are hollow

spaces in the cortex of stems and roots that allow $O_2$ to move from the atmosphere to the rhizosphere, an adaptation for wetland or inundated environments. $CH_4$ and other gases produced in the rhizosphere can potentially move to the atmosphere through these tissues and out lenticels, which have been studied as gas transport pathways in black alder (*Alnus glutinosa*) (Schroder, 1989; Buchel and Grosse, 1990). It has been proposed that $CH_4$ can be transported through these structures (Rusch and Rennenberg, 1998). Gas transport through plant tissues is usually suggested to be diffusive in current literature, whether

for emergent aquatic plants (Chanton, 2005) or upland or riparian trees (Machacova et al., 2013; Pangala et al., 2014).

Temperature, among other factors, is known as a significant factor in the control of wetland $CH_4$ emission (Whalen, 2005) and it is believed that increased soil temperatures lead to increased $CH_4$ flux from rice fields (Khalil et al., 1998b). However, it is unknown how temperature specifically affects transport mechanisms. Whether diffusive or otherwise, this relationship is important since wetland emissions are expected to increase in a warmer, wetter climate (Ciais et al., 2013). Faster transport

of $CH_4$ out of the soil by trees leaves less time for its oxidation by methanotrophic bacteria; likewise, a slower transport time means more $CH_4$ will be oxidized in the soil and less will be emitted overall. Our study experimentally probes whether diffusion through tree stems (a slower process) or transpiration (a faster route) are likely to be significant transport mechanisms for $CH_4$. These transport mechanisms will themselves be affected differently by temperature; therefore it is important to know what mechanisms are at work as well as how those mechanisms are changed with temperature.

In this study we have investigated the transport mechanisms of $CH_4$ through *Populus trichocarpa* or black cottonwood. By measuring $CH_4$ flux and the isotopic fractionation of emitted $cH_4$, we examine the possible transport mechanisms (e.g. bluk flow, diffusion) and rule out those that are not supported by our measurements. We also examine the variation of flux and isotopic fractionation with temperature, which gives further insight into the likely modes of $CH_4$ transport and how those modes would be affected by temperature change.

## 2   Methods

### 2.1   Growth Conditions and Canopy Flux Measurements

Native to the Pacific Northwest, *Populus trichocarpa*, a poplar species also known as black cottonwood, can be found in floodplains and areas where inundation occurs. This species is used as a landfill cover (McBain et al., 2004). Since it is flood-tolerant, it can be grown hydroponically in order to isolate tree transport mechanisms from the complications of plant-soil

interactions. Cuttings from wild black cottonwood were taken from the Sandy River Delta floodplain in the Columbia River Gorge, Troutdale, Oregon. By taking cuttings and growing roots hydroponically, roots can be kept from any exposure to soil methanogenic or methanotrophic bacteria. 7.5 liter plastic buckets with tight-fitting lids were filled to 5.7 liters with a modified Hoagland's solution and two 3 cm diameter holes were cut in the plastic lids, one on each half of the lid. One cutting was



placed in each hole and held in place by a foam stopper, then the lid was secured to the bucket. The nutrient solution was changed every ten days. Every other day the pH was checked and sodium hydroxide added to the solution if the pH was found to be lower than 5.8. In case of contamination by methanogens of the growing buckets, root water was tested just before being changed. $CH_4$ was found at just above the detection limits of the gas chromatographs, five orders of magnitude less than the

$CH_4$ concentrations during the experiments described below. The small amount of $CH_4$ detected may have been dissolved from ambient air; production of $CH_4$ in the root zone of the growing buckets was concluded to be unlikely.

Saplings were cultivated for at least four months under greenhouse conditions with a photoperiod of 16 hours before experimentation began. Natural light was used unless the intensity dropped below a pre-set PPFD, in which case greenhouse lighting was set to activate. Light levels measured at leaf surfaces at the time of an experiment averaged 72 $\mu$moles light m$^{-2}$ s$^{-1}$ but

ranged from 3 to 530 $\mu$moles light m$^{-2}$ s$^{-1}$ (LI-250A Light Meter, LI-COR Biosciences). Trees grew during the experimentation period and ranged in height from about 75 cm to 1 meter above the bucket lid level of 26 cm. Leaves on each tree were photographed once and leaf area calculated from pixel counts. A correlation of leaf area versus leaf count was done to estimate the leaf area of each tree based on leaf count at the time of each flux experiment (r$^2$ = 0.79). However, as this was an estimate it was not used for reporting flux per leaf area.

To perform an experiment, one tree was removed from the growing vessel and the tree's roots placed in an acrylic root chamber based on a design used by Rusch and Rennenberg (1998), as depicted in Figure (1). The chamber was filled with deionized water through which natural gas had been bubbled until a high concentration of $CH_4$ was reached. The concentration of $CH_4$ in the root chamber was similar to that found in flooded tubs of soil in greenhouse experiments where no plant was present (unpublished data). These concentrations averaged 0.7 $\mu$moles $CH_4$ mL$^{-1}$ water (standard deviation $\sim$ 0.1 $\mu$moles

$CH_4$ mL$^{-1}$ water), which is in the range of $CH_4$ water concentrations in a $CH_4$ flux study done on hydroponically grown rice (Yao et al., 2000). The root chamber was closed around the tree stem with an o-ring between the chamber and chamber lid, then sealed around the stem with modeling clay to separate the water and roots from the ambient air. Electrical tape was used to cover the seams of the root chamber lid. Aluminum foil covered the transparent root chamber to protect the roots from ambient light. After enclosing the tree in the root chamber, the system was left in the greenhouse for at least three hours before samples

were taken in the afternoon or early evening.

The water level in the root chamber was marked once the tree roots were enclosed and the chamber sealed. Just before an experiment, the water level was refilled to the initial marked level. The amount of water used to refill the root chamber was taken as a measurement of water transpired by the tree. Also before the experiment, leaf temperature, ambient temperature and relative humidity were recorded. The average ambient temperature was 18 $\pm$ 2 $^o$C for these experiments; a set of trials

with temperature variation are described in the next section. For sampling, a 100 liter tedlar bag was used to enclose the entire canopy of the tree. Since the bottom of the bag had been cut off and some of the bag material wrapped around the tree stem to seal the canopy, the net volume of the bag was approximately 90 liters. Any variation in $CH_4$ concentration due to these minor volume variations was later calculated to be dwarfed by the variability observed in flux data. 20 mL of canopy air was sampled with a syringe from a septum on the tedlar bag every five minutes for a half hour, beginning as soon as the tedlar bag

was closed around the base of the stem. 10 mL water samples were taken every ten minutes using a syringe from a sampling



port on the root chamber. 20 mL ambient greenhouse air samples were taken just after the bag was closed and just before it was removed. All samples were then analyzed for $CH_4$ concentration on an Agilent model 6890 gas chromatograph (GC) with a flame ionization detector (FID)(Khalil et al., 1998a). The rate of $CH_4$ accumulation in the chamber was analyzed by linear regression to obtain the net $CH_4$ flux. Water samples were mixed with 20 mL $N_2$ and agitated by shaking for five minutes.

The water was then expelled and the $CH_4$ concentration determined by measurement on the GC-FID. The dissolved $CH_4$ concentration of the water was calculated by the method described in Lu et al. (2000).

Canopy $CH_4$ flux from each of eight trees was measured twice during November and December of 2010, and a third and fourth time on six of the original trees in March and April of 2011. Some growth occured between the fall and spring trials. Leaf area was measured between the 2010 and 2011 set of flux measurements and calculated non-destructively as described

above. Stem diameters were also recorded. Leaf areas ranged from 0.09 $m^2$ to 0.49 $m^2$ with an average of 0.34 $m^2$. Leaf counts ranged from 125 to 424 leaves per tree with an average of 289 per tree. The correlation between leaf count and leaf area yielded a determination coefficient of $r^2 = 0.79$.

Stem diameters were measured also. The diameter measurement was taken at 22 cm from the bottom of the original cutting, which was approximately the height where root growth from the cutting began. Stem diameters ranged from 0.010 m to 0.016

15    m with an average of 0.013 m.

## 2.2    Temperature Variation Experiment

For the temperature experiment, black cottonwood was grown hydroponically as previously described. Five trees were used, taken as cuttings from one black cottonwood tree growing in the Sandy River Delta, Troutdale, Oregon, on September $19^{th}$, 2011. Experimentation began on January $30^{th}$, 2012, and ran for just over three weeks. Each week, the trees were moved to a

greenhouse room with a different temperature while other environmental factors such as day length were kept constant. Leaf counts ranged from 131 to 500 leaves per tree with an average of 329 per tree; this is comparable to the trees used in the previous set of experiments as their sizes were similar. The average day-time temperature was 22.3 $^oC$ the first week, 25.2 $^oC$ the second week, and 17.1 $^oC$ the third week (compared to an average ambient temperature of $18 \pm 2$ $^oC$ for the experiments described in the previous section that measured canopy flux without temperature variation). Natural gas was bubbled through

deionized water in the root chamber for approximately 45 minutes before the start of each experiment. The root chamber was then placed on a scale (Mettler Toledo, New Classic MF, model MS1200 1L) in the greenhouse and weighed. After moving the tree from the growing container to the root chamber, the difference in weight was recorded as tree weight. Ambient temperature and relative humidity were recorded once the tree was secured in the root chamber and during the experiment. Canopy samples and root water samples were taken and processed in the same manner as described in the previous section for measuring canopy

flux without temperature variation.

## 2.3    Isotopic Measurements

In the canopy flux experiments outlined above, both with temperature variation and without, carbon isotopic composition of $CH_4$ was measured. Samples of canopy $CH_4$ flux were taken at 30 minutes into each experiment for $\delta^{13}C$ analysis. Two 60 mL



syringes were taken from the canopy bag, two from the ambient air, and 10 mL water was taken from the root chamber to be mixed with 50 mL $N_2$ and processed as described above. These samples were injected into stoppered 45 mL glass storage vials. Syringe samples from these vials were measured by continuous-flow gas chromatography-isotope ratio mass spectrometry on a Thermo Scientific Delta V Advantage IRMS using the method described in Rice et al. (2001). Values for the canopy $CH_4$ were corrected using the ambient greenhouse samples.

By convention the ratio of heavy to light isotopes of $CH_4$ is expressed with the delta notation as

$$\delta^{13}C\text{\textperthousand} = \left[\frac{R_{samp}}{R_{std}} - 1\right] \cdot 1000 \tag{1}$$

where $R_{samp} = {}^{13}C/{}^{12}C$ and $R_{std}$ is the known ${}^{13}C/{}^{12}C$ ratio of Vienna PeeDee belemnite limestone (VPDB) (Gonfiantini et al., 1995).

Discrimination for or against ${}^{13}C$ in a particular process can be expressed by the ratio of ${}^{13}C$ to ${}^{12}C$ for the process, or $\alpha = \frac{R_A}{R_B}$ where $R_A$ is the ${}^{13}C/{}^{12}C$ ratio prior to the process and $R_B$ is the ratio afterwards. The process may be a chemical reaction, dissolution from water into a gas phase, or diffusion along a gradient. $\alpha$ is known as the fractionation factor. Also convenient is the expression $\epsilon = (\alpha - 1) \cdot 1000$ Using equation (1) this can be written as

$$\epsilon = \left[\frac{\delta_A + 1000}{\delta_B + 1000} - 1\right] \cdot 1000 \tag{2}$$

It can be shown (Hoefs, 2004) that for small values of $\epsilon$, $\epsilon \sim \delta_A - \delta_B$.

### 2.4 Stem Experiments

In the interest of separating stem emission from any potential emission occuring from the leaves, two acrylic cylindrical stem cuvettes (20cm in length, 5.8 cm radius) were used to measure flux exclusively emitted from the stem, based on a design used by Rusch and Rennenberg (1998) and depicted in Figure (2). These cuvettes were attached around a section of stem 11.7 cm in length, or 23 cm if both stem chambers are used. Experiments with the stem cuvettes were identical to those outlined in the section for canopy flux measurements, except instead of the tedlar bag, the stem cuvettes were attached for a half hour around the tree stem while flux samples were taken every five minutes from each cuvette. The tree was secured in the root chamber for a minimum of 190 minutes and a maximum of 370 minutes before measurements were taken. Cuvettes were placed along the main stem of the tree, one above the other. Water concentration samples were taken as they were for the canopy flux experiments. A total of four trials were conducted, two of which were at different times with the same tree.

## 3 Results

### 3.1 Canopy Flux

Concentration of $CH_4$ in the tedlar bag versus time was graphed for all trials. Of the 28 canopy flux trials without temperature variation performed 2010-2011, 18 had an r-squared value over 0.95 and three had an r-squared value between 0.9 and 0.95.



Statistical analysis includes only the 18 trials with r-squared values over 0.95, leaving out trials where inconsistent factors may have altered fluxes. Of the 17 trials conducted in the 2012 temperature variation experiment, the 16 trials with an r-squared above 0.95 are included in the analysis below. These results indicate that for the canopy, emissions are normally linear over the half-hour sampling period. Flux is reported for the entire canopy, that is, the total change in $CH_4$ concentration in the sampling

chamber. The average temperature for the 18 trials in 2010-2011 was 18.3 $^oC$; the average flux was $3.0 \pm 2.6$ $\mu g$ $CH_4$ $min^{-1}$. In the 2012 temperature variation experiment, average daytime temperature the first week was 22.3 $^oC$, 25.2 $^oC$ the second week, and 17.1 $^oC$ the third week. The average flux during the 2012 temperature experiment was $2.5 \pm 1.7$ $\mu g$ $CH_4$ $min^{-1}$. The average flux for the warmest temperature, 25.2 $^oC$, was $3.2 \pm 1.4$ $\mu g$ $CH_4$ $min^{-1}$; the average flux at the next highest temperature, 22.3 $^oC$, was $3.2 \pm 2.2$ $\mu g$ $CH_4$ $min^{-1}$; at the coolest temperature of 17.1 $^oC$, the average flux was $1.6 \pm 0.8$ $\mu g$

$CH_4$ $min^{-1}$.

Although the constant-temperature experiments were conducted at a different time than the temperature-varying experiments, since a relationship between temperature and flux was sought the data from these experiments were combined. As described in the Methods section, growth conditions were the same in each case with the exception of temperature variation. When analyzed, the fluxes were not normally distributed. Therefore, the nonparametric Kruskal-Wallis test was performed in

place of ANOVA. Fluxes were binned into three groups, the first ranging from $16^o$ C to $19.1^o$ C, the second from $20.4^o$ C to $22.1^o$ C, and the third from $24.4^o$ C to $28.7^o$ C. Fluxes were found to vary significantly with temperature ($p = 0.0012$). A linear (robust) regression of $CH_4$ flux versus temperature is shown in Figure (3) ($R^2 = 0.61$; the p-values of the coefficients f(x) = $2.53x10^{-7}$ x - $2.67x10^{-6}$ are $7.0x10^{-4}$ and $0.60x10^{-1}$).

### 3.1.1 Isotopic Data

Average root water $\delta^{13}C$ was -36.7‰ and did not vary significantly (95% C.I. (-36.2‰, -37.2‰)). Compared to the isotopic composition of root water $CH_4$, $\delta^{13}C$ values of canopy $CH_4$ were depleted on average by $7.8 \pm 4.4$ ‰. These were binned into the same three temperature groups as the flux data given above; results are given in Table (1). The warmest temperatures (24.4-28.7 $^oC$) resulted in an epsilon of $2.8 \pm 4.7$‰; midrange temperatures (20.4- 22.1 $^oC$) produced an epsilon of $7.5 \pm 3.1$ ‰; the coolest temperatures (16-19.1 $^oC$) resulted in an epsilon of $10.2 \pm 3.2$ ‰.

Epsilon values of the emitted $CH_4$ have been graphed as a function of flux in Figure (4) and as a function of temperature in Figure (5). The value of $\epsilon$ decreases with increasing flux and increasing temperature: the root water values are stable from saturating the root chamber with $CH_4$, and canopy $\delta^{13}C$ values become more enriched with increasing temperature and increasing flux. Both flux and temperature have a statistically significant correlation with epsilon ($R^2$ for epsilon versus flux is 0.37; the p-values of the coefficients f(x) = $-1.64x10^6$ x + 11.8 are $7.12x10^{-6}$ and $5.33x10^{-12}$; $R^2$ for epsilon versus temperature is also

0.37; the p-values of the coefficients f(x) = -0.88 x + 25.1 are $4.51x10^{-5}$ and $2.57x10^{-7}$).

### 3.2 Stem Flux

Results from the stem chamber experiments are summarized in Table (2). The lower stem chamber is referred to as S1 and the upper chamber as S2. Placement of the stem chambers (lower side) ranged from 0.03 m to 0.46 m along the main tree stem.





The raw flux is given from each stem chamber: fluxes ranged from 0.015 to 1.7 $\mu$g $CH_4$ min$^{-1}$ with an average of 0.65 $\mu$g $CH_4$ min$^{-1}$. Fluxes per enclosed stem area are simply the raw flux divided by the average stem area enclosed by the chamber (0.0058 m$^2$ for the first trial, 0.0048 m$^2$ for the following three). The estimated total stem flux is calculated by dividing the total main tree stem length by the length enclosed by the two stem chambers, then multiplying by the sum of the two stem chamber

fluxes. Obviously this calculation assumes that flux along the entire tree stem is consistent and represented by that measured in the two chambers; the relevance of these values will be taken up in the discussion. Transpiration, based on water loss from the root chamber, is also reported.

## 4  Discussion

As a common species in wetland environments that also grows well under non-inundated conditions, it is assumed that transport

mechanisms in poplar will be representative of other tree species in similar environments. The versatility of poplar makes it ideal for transport studies in this sense. Although experiments were conducted on trees grown in a hydroponic environment, the physiological adaptations of poplar expressed under inundation in the field should be same; therefore the transport mechanisms themselves should not differ in essence. Hydroponically grown rice has been used for $CH_4$ transport studies where greater control was necessary than what was possible in the field, yet the influence of physical characteristics of the plant or temperature

on $CH_4$ transport were studied (Hosono and Nouchi, 1997; Yao et al., 2000).

Regardless of the mechanisms, trees having the capability of moving $CH_4$ from the soil to the atmosphere will do so when $CH_4$ is present in the root zone, as demonstrated by the fact that in all experiments where $CH_4$ was present in the root zone, flux was measured from the canopy. Although it has been suggested that the developmental stage of a tree may play a role in the ability of a tree to transport $CH_4$ (Pangala et al., 2014), the specific $CH_4$ transport mechanisms themselves are not expected

to be altered by more mature tree growth. Therefore, the mechanisms described here should be applicable qualitatively to more mature trees, even though there may be quantitative shifts in $CH_4$ flux.

For the more thoroughly studied rice plant, correlations between morphophysiological and anatomical characteristics of rice plants and $CH_4$ emission exist (Nouchi et al., 1990; Das and Baruah, 2008). Although a large amount of $CH_4$ has been shown to come through emergent wetland plants (Shannon et al., 1996), physiological correlations for tree emission of $CH_4$ have

been studied very little and the role that woody vegetation has in wetland emission of $CH_4$ is still poorly understood (Vann and Megonigal, 2003; Terazawa et al., 2007). It should be mentioned that more recently, Pangala et al. (2014) did find a strong correlation between lenticel density and $CH_4$ flux from stems in the tree species *Alnus glutinosa*. It is clear from the data here that temperature and flux, $\delta^{13}C$ and flux and $\delta^{13}C$ and temperature are significantly related. Further study is needed to establish rigorous links between $CH_4$ flux from trees with physiological parameters.

Little evidence was found to indicate leaf flux in general. Calculations dividing $CH_4$ flux by estimated leaf area indicated that any leaf emission would be below detection limits; regardless, twice leaf emissions were tested using a leaf cuvette instead of a canopy bag under the same experimental conditions as described above. No emission was measured. $CH_4$ emission measured in the stem experiment and scaled to the stem area for the tree yields a total $CH_4$ flux value in the range of measured fluxes for





the entire canopy. The average flux for the 18 canopy emission trials in 2010-2011 was $3.0 \pm 2.6$ $\mu$g CH$_4$ min$^{-1}$. From Table (2), the estimated total stem fluxes from the main stem of the tree based on stem chamber measurements were 3.1, 3.8, and 12 $\mu$g CH$_4$ min$^{-1}$. This suggests that a considerable amount of the total CH$_4$ emitted exits from stems after traveling through the tree, possibly via air-filled aerenchyma tissue. A decrease in flux was observed with height of the stem chamber along the

stem, consistent with findings by Rusch and Rennenberg (1998); Terazawa et al. (2007); Pangala et al. (2013). In the case of Rusch and Rennenberg (1998), this was taken as evidence for the major transport of CH$_4$ to be by diffusion. While the stem pathway does appear to dominate, the exact mechanisms of transport may or may not include diffusion as discussed below.

An analysis of the possible physical pathways of CH$_4$ through the tree is helpful in interpreting the isotopic fractionation of CH$_4$ observed from canopy measurements. CH$_4$ entering the root system with water can move apoplastically, or without

traversing cellular membranes. A membrane barrier at the root endodermis must be crossed before the xylem tissue is entered, which is the main bulk-flow water transport system through roots and stems of trees (Tyree and Ewers, 1991). This xylem pathway is driven by a hydrostatic pressure gradient, as opposed to the water potential that moves water from cell to cell across membrane barriers (Tyree and Ewers, 1991). The bulk flow of water causes no isotopic fractionation. Therefore, CH$_4$ transported by transpiration would show an isotopic fractionation due to dissolution from water to gas and from the crossing

of two cell membrane barriers at the root endodermis. Experimentation is needed to know the isotopic fractionation due to membrane transport.

Alternatively, CH$_4$ may enter the roots without ever being drawn into the xylem tissue, instead entering air-filled aerenchyma tissue connected to lenticels along the tree's stems. This pathway is believed to be the dominant pathway for CH$_4$ flux from rice and some wetland plants (Denier Van Der Gon and Van Breemen, 1993; Chanton, 2005). Isotopic fractionation would

occur due to dissolution from water to gas and from any concentration-gradient driven diffusion between the root zone and the atmosphere. Transfer of CH$_4$ from water to air leads to an isotopic fractionation on the order of -0.8‰(Knox et al., 1992), an order of magnitude smaller than the fractionation measured here.

Molecular diffusion is described by Fick's first law, where molecules move along a concentration gradient by random collisions:

$$J = D\frac{dC}{dz} \tag{3}$$

$J$ is the flux in mass per area per time, $D$ is the diffusion coefficient in length$^2$ per time, $C$ is the concentration and $z$ is distance, and d$C$/d$z$ is the concentration gradient. For a gas moving along a concentration gradient through air, the diffusion coefficient is related to the molecular weights of the constituents:

$$D_{12} \propto \left[ \frac{(M_1 + M_2)}{M_1 M_2} \right]^{1/2} \tag{4}$$

where $M_1$ is taken to be the molecular weight of air and $M_2$ for the molecular weight of CH$_4$ (Mason and Marrero, 1970). The expected fractionation for CH$_4$ diffusing in air is found from the ratio of the diffusion constants for $^{12}$CH$_4$ and $^{13}$CH$_4$. Using equation (4) and taking the molecular weight of air to be 28.9 g mol$^{-1}$, the $\alpha$ for diffusion is 1.019. The resulting isotopic fractionation due to molecular diffusion, therefore, is expected to be -19‰(Chanton, 2005).




The isotopic fractionations of $\sim$3 - 10‰ measured here indicate that $CH_4$ transported through the tree follows neither a purely bulk flow pathway nor a purely diffusional one, as has been suggested for rice and other wetland plants. The value can be interpreted to indicate two or more mechanisms of transport, possibly some combination of movement through membrane barriers, bulk flow, and diffusion. However, although gas transport through aerenchyma is usually assumed to be diffusive

(Chanton, 2005; Garnet et al., 2005; Machacova et al., 2013; Pangala et al., 2014), this work strongly suggests that molecular diffusion is not the primary transport mechansim. It should be noted that these fractionation values reflect the total combination of transport mechanisms through the tree: if any $CH_4$ does exit from other parts of the tree while most exits through the stem, the isotopic fractionation measured is from all $CH_4$ emitted. In other words, the values give indications about the combination of mechanisms along the pathways but does not discriminate between the pathways themselves.

The significant relationship between epsilon and canopy $CH_4$ flux (Figure (4)) can be explained by different transport mechanisms dominating depending on the amount of $CH_4$ emitted. Given at least a small amount of $CH_4$ in the root zone, a low flux occurs that is dominated by a particular transport mechanism or mechanisms with a larger isotopic fractionation. When various ecophysiological parameters, such as temperature, combine to result in a high flux, more $CH_4$ is emitted by a different transport mechanism or mechanisms with a correspondingly smaller fractionation as shown in Figure (5). This explains the

decrease in epsilon with both flux and temperature; however, given the current data it is not possible to separate the influence of these two factors as drivers of epsilon.

## 5 Conclusions

Based on this and previous work, multiple mechanisms appear to be likely for $CH_4$ transport through trees. Most, if not all, $CH_4$ is transported through tree stems and not with transpiration. At least two mechanisms may be at work in this process:

one dominant at low flux (and low temperature) leading to a substantial fractionation between root water and emitted $CH_4$, and another mechanism dominant at higher flux (and higher temperature) with little or no fractionation. Molecular diffusion cannnot be treated as the sole transport mechanism responsible for plant transport of $CH_4$; isotopic data suggests it is not even a dominating mode of transport. As for $CH_4$ emissions from upland soils, because there is evidence against significant $CH_4$ transport through leaves with transpiration and stem emission is not supported in the literature for those conditions, it appears

unlikely that $CH_4$ produced in the soil would be emitted in large quantities before it was oxidized.

Accurately predicting emissions feedbacks in the soil-tree-atmosphere system will depend on knowing the rate at which a particular transport mechanism changes for a given environmental parameter, e.g., temperature. An increase in $CH_4$ flux corresponding to temperature may be linked to a temperature-dependent mechanism within the tree, as suggested here by the fractionation data. While one part of the system may experience a strong positive feedback, such as increased $CH_4$ production

in the soil with temperature, the feedback on transport mechanisms of $CH_4$ through trees could mitigate the overall effect.



*Acknowledgements.* This work was funded by the Office of Science (BER), U.S. Department of Energy, grant number DE-FG02-08ER64515 and by NASA / Oregon Space Grant Consortium, grants NNG05GJ85H and NNX10AK68H. The authors would like to thank D. Teama, M. Shearer and D. Matarese for laboratory assistance. Data are available upon request.



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



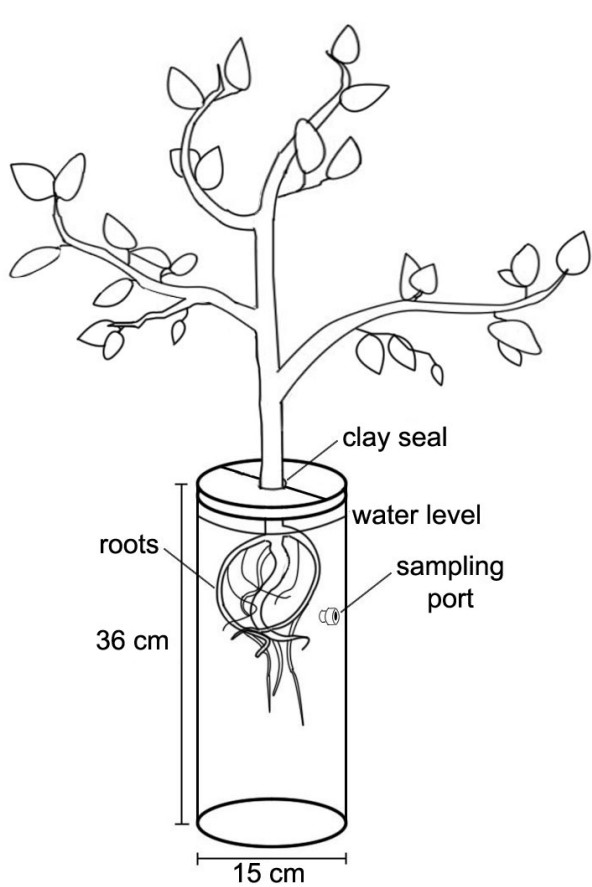

**Figure 1.** Depiction of experimental setup with dimensions. Four springs (not shown) are evenly spaced around the chamber lid, attaching to the chamber body and holding the lid down by tension. Modeling clay is used to seal between the chamber and the tree stem; electrical tape is run along the seams of the lid to prevent any leaking.




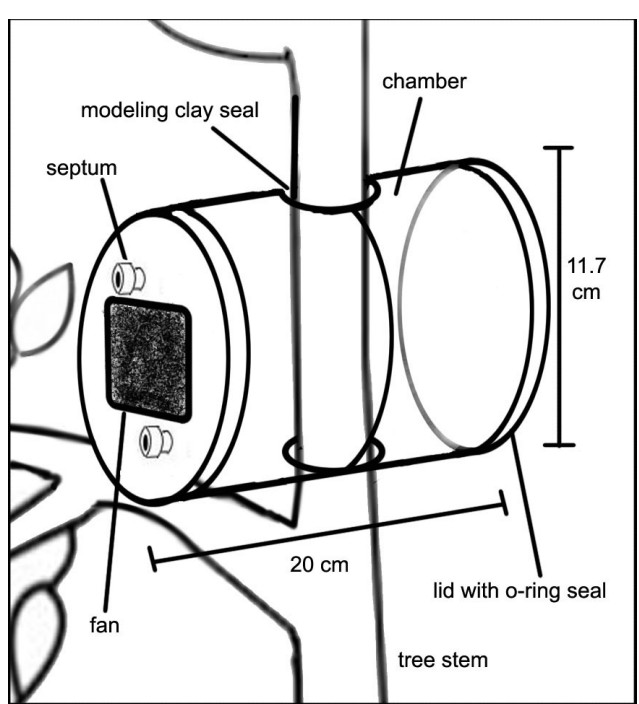

**Figure 2.** Depiction of stem chamber attached to tree. Stem chamber design is based on (Rusch and Rennenberg, 1998). Actual lid pieces are square bases into which the cylindrical halves of the chamber fit, sealed with an o-ring. Not depicted are the four springs, one at each corner of the square lid base, that attach to the opposite lid piece and hold the chamber together by tension. Modeling clay is used to seal between the chamber and the tree stem; electrical tape is run along the middle seam of the chamber to prevent any leaking. Two wires running from the chamber fan and sealed through the lid with glue connect to a 12-volt battery external to the chamber.





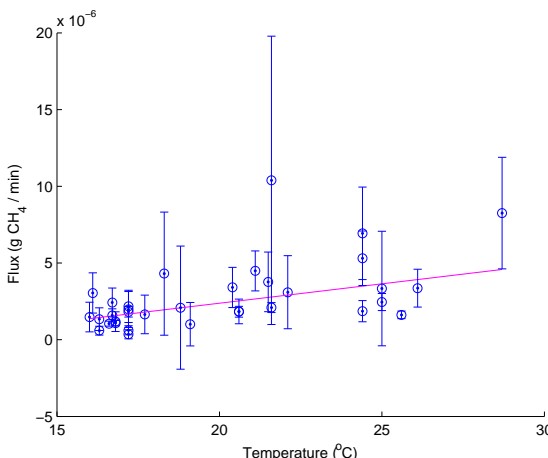

**Figure 3.** $CH_4$ flux emitted from the entire canopy versus temperature for 2010-11 experiments and 2012 temperature variation experiments, n=34. Regression coefficients f(x) = $2.53 \times 10^{-7}$ x - $2.67 \times 10^{-6}$; $R^2 = 0.61$; SE of coefficients $6.73 \times 10^{-8}$, $1.37 \times 10^{-6}$; p-value of coefficients $7.0 \times 10^{-4}$, $0.60 \times 10^{-1}$. Error bars given by the standard error of each flux measurement.

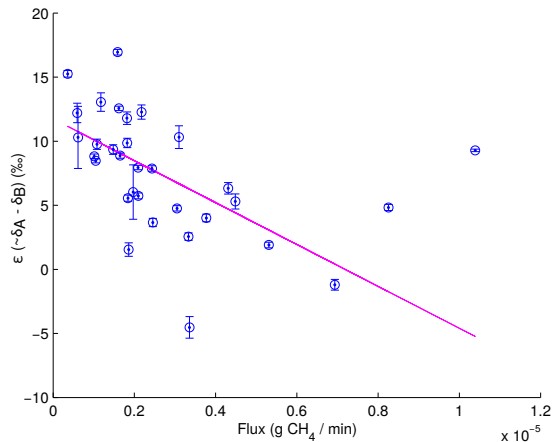

**Figure 4.** Regression of epsilon ($\epsilon$) versus $CH_4$ flux for the whole canopy (n=32) using linear (robust) techniques. Epsilon is the difference between the $\delta^{13}C$ of the root zone water ($\delta_A$) and the canopy flux ($\delta_B$). Regression coefficients f(x) = $-1.64 \times 10^6$ x + 11.8; $R^2 = 0.37$; SE of coefficients $3.02 \times 10^5$, 1.07; p-value of coefficients $7.12 \times 10^{-6}$, $5.33 \times 10^{-12}$. Error bars represent the standard error for each value of epsilon.





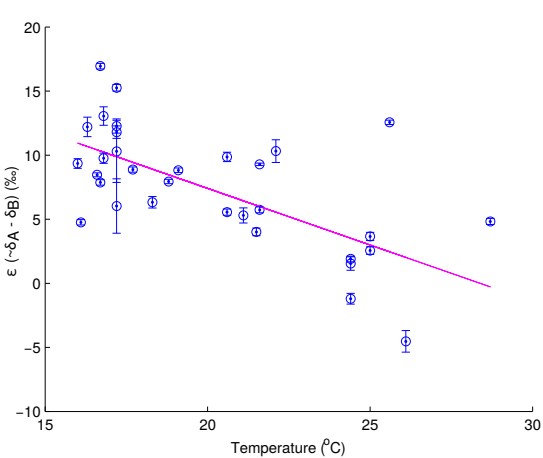

**Figure 5.** Regression of epsilon versus temperature (n=32) using linear (robust) techniques. Epsilon is the difference between the $\delta^{13}C$ of the root zone water ($\delta_A$) and the canopy flux ($\delta_B$). Regression coefficients f(x) = -0.88 x + 25.1; $R^2 = 0.37$; SE of coefficients 0.19, 3.79; p-value of coefficients $4.51 \times 10^{-5}$, $2.57 \times 10^{-7}$. Error bars represent the standard error for each value of epsilon.



**Table 1.** $\delta^{13}$C Values binned by temperature for the 2010-11 canopy flux experiments and 2012 canopy flux experiments with temperature variation.

| temperature °C | root water avg. $\delta^{13}$C | N | SD | 95% C.I. | canopy avg. $\delta^{13}$C | N | SD | 95% C.I. | epsilon |
|---|---|---|---|---|---|---|---|---|---|
| 16-19.1 | -36.5 | 16 | 0.73 | (-36.1, -36.9) | -46.3 | 17 | 3.1 | (-44.8, -47.7) | 10.2 |
| 20.4-22.1 | -36.7 | 7 | 1.2 | (-35.8, -37.6) | -43.8 | 7 | 2.8 | (-41.8, -45.9) | 7.5 |
| 24.4-28.7 | -37.1 | 8 | 2.2 | (-35.6, -38.7) | -39.8 | 8 | 4.2 | (-36.9, -42.7) | 2.8 |

**Table 2.** Stem fluxes. Stem chamber height refers to the height of the lower side of the stem chamber along the main tree stem, above the top of the root chamber. Main stem heights were 0.90 m, 0.49 m for the second and third trials, and 1.62 m for the final trial. The second and third trials were performed on the same tree. *In the second stem experiment, the clay seal over the root chamber came in contact with the clay seal over the lower part of S1, creating a leak into the stem chamber and an artificially high flux.

| stem chamber | stem chamber height (cm) | flux ($\mu$g CH$_4$ min$^{-1}$) | flux per enclosed stem area ($\mu$g CH$_4$ min$^{-1}$ m$^{-2}$) | est. total stem flux ($\mu$g CH$_4$ min$^{-1}$) | transpiration (mmol H$_2$O s$^{-1}$) |
|---|---|---|---|---|---|
| trial 1 - S1 | 0.30 | 0.57 | 97 | 3.1 | 0.24 |
| trial 1 - S2 | 0.46 | 0.23 | 39 | | |
| trial 2 - S1 | 0.03 | error* | * | * | 0.73 |
| trial 2 - S2 | 0.19 | 0.18 | 37 | | |
| trial 3 - S1 | 0.03 | 1.6 | 340 | 3.8 | 0.23 |
| trial 3 - S2 | 0.17 | 0.22 | 46 | | |
| trial 4 - S1 | 0.04 | 1.7 | 360 | 12 | 0.30 |
| trial 4 - S2 | 0.30 | 0.015 | 3.1 | | |