# Peer review of "Mechanisms of methane transport through *Populus trichocarpa"

_Biogeosciences, 2016_

## Referee Comment (RC1) · Anonymous Referee #1 · 26 Apr 2016

MS #: Biogeoscience Discussion BG-2016-60

Title: Mechanisms of methane transport through Populus trichocarpa

1. General comments

Plant-mediated transport is an important emission pathway of soil-born methane (CH4) to the atmosphere in natural wetlands and rice paddies. In these ecosystems, several different modes of CH4 transport have been reported from aquatic and herbaceous plants, which includes both diffusion and convective flows, or gaseous transport through internal air spaces (e.g., aerenchyma) in plant body and dissolved form in transpirational stream. For woody plants, however, despite the fact that some tree species, especially those adapted to flooded soil conditions, can act a conduit for CH4 transport from soil to the atmosphere, mechanisms of CH4 transport through tree bodies have

not been well-understood. In this context, the aims of this experimental work tackling the CH4 transport mechanisms within a tree body is quite relevant.

I regret to comment, however, that there are not a few problems mainly in the methodology of this work; some of which would be basic or fundamental problems which could draw wrong conclusions. Details of the problems are described below.

2. Specific comments

(1) Measurement of CH4 emission rate

i) In this study, CH4 emissions were measured on an individual plant basis, and the emission rates reported in this paper are the values per plant, instead of those per unit area of stem surface or leaves. Since the tree saplings used for the flux measurements differed significantly in size (P.4, L.10 – 15), the emission rates must have been influenced primarily by plant size. Nevertheless, the authors simply compared those values of CH4 emission rate in relation to the temperature (Fig. 3) or isotopic signature of CH4 (Fig.4). I wonder why such analyses and discussion are possible.

ii) The authors used a 100-L Tedlar bag to enclose the entire aboveground part of each sapling for the CH4 flux measurement. However, the actual internal volume of each Tedlar bag, which certainly is an essential parameter for the calculation of CH4 flux, seemed to be unknown, because the bottom of the bag was cut off and the plant was wrapped by the bag and sealed again. The authors mentioned that the net volume of the bag was "approximately 90 L" (P.3, L.32), but there is no description on how they measured its actual volume.

(2) Temperature variation experiment

i) Effects of temperature on CH4 transport within a plant body or related physiological properties of plants should be tested with direct measurements of temperature to which the plant are actually exposed. However, in the temperature variation experiment of this study, the inside temperature of each Tedlar bag for flux measurement was not

recorded, and the water temperature in the root chamber neither. Although the ambient temperature of the greenhouse during the experiment was measured (P.4, L.27 – 28), we cannot rule out the possibility of substantial differences in temperature between ambient air and inside air of the bag, or between ambient air and water surrounding the root system of plants.

ii) When the authors analyzed the temperature effects on the $CH_4$ flux and isotopic fractionation, they pooled the data obtained from two different sets of measurements, which were conducted at different season: one from November 2010 to April 2011, and another from January to February 2012. If we take account of the potential influences of phenological difference on physiological and anatomical traits of trees, it would be quite difficult to combine those two set of data to analyze the temperature effects, especially for deciduous tree species like Populus trichocarpa. Furthermore, I wonder whether the origins of tree saplings (cuttings) used for these two set of experiments were same plant or not. If the cuttings were taken from different individual plants, it would be more difficult to combine the data because of the possibility of genetical differences in physiology and anatomy, and those responses to temperature.

(3) Estimation of total stem flux

i) The authors tried to estimate the total flux of $CH_4$ emitted from the stem surface using a stem chamber (Fig. 2). The results showed that stem $CH_4$ emission rate was much lower at the higher position on a stem (Table 2), showing the similar tendencies reported from some other studies (e.g., Rusch and Rennenberg 1998, Terazawa et al. 2007, Pangala et al. 2013, 2014). Nevertheless, the authors did not take this decreasing tendency of $CH_4$ flux with stem height into account at all, and "the estimated total stem flux is calculated by dividing the total main tree stem length by the length enclosed by the two stem chambers, then multiplying by the sum of the two stem chamber fluxes" (P.7, L.3 – 6). There is no convincing explanation regarding the appropriateness of the estimation.

(4) Other points

i) There is no description on the leaf cuvette in the section of methods, while it is mentioned in the discussion (P.7, L.31).

ii) Each value in the Fig.3 is supposed to represent the flux value obtained from one single measurement for a tree. I wonder why each value has the range of standard error shown by the error bar.

---

## Referee Comment (RC2) · Anonymous Referee #2 · 29 Apr 2016

General comments Further understanding of the role of woody plants as a pathway for methane emissions from wetlands is necessary. Wetlands contribute significant methane to the atmosphere and the contribution from woody stems is unknown. The mechanisms in which soil methane can pass through woody stems are addressed within this research. The focus was to define the ratio of diffuse and sap flux driven methane emissions from woody vegetation and understand the role of temperature on these pathways.

There are some shortcomings with this work that could influence the authors to draw accurate conclusions. For example, confounding variables like vapor pressure deficit in the Tedlar bag compared to ambient vapor pressure deficit could limit the ability to measure transpiration's influence on stem methane emissions.

[Figure]

Specific comments

i. Lenticels are described and referenced as a major driver of woody stem methane emissions. The authors do not indicate any measure of the quantity of lenticels.

ii. The number of lenticels increases due to an elevated water table and the associated stress on the tree, were water levels fluctuated to promote lenticel growth on sample saplings?

iii. The authors present the fact that black cottonwood is frequently used as a cover species for landfills (page 2 line 28: "This species is used as a landfill cover"). The implication that woody vegetation can allow gases to pass from the soil to the atmosphere is not discussed later in the paper. A connection between the findings and landfill pollutant emissions would be beneficial in the discussion section.

iv. In the Methods section the authors say that the hydroponically cultivated sample trees were grown in a modified Hougland's solution; they should indicate how it was modified or give a reference for this step.

v. The authors should clarify whether or not the air chambers were tested for airtightness.

vi. The authors do not state if there was a method to ensure appropriate mixing of air within chambers or Tedlar bags.

---

## Author Comment (AC1) · 6 May 2016

Thank you for your comments. The reply to each comment is enumerated below:

(1) (i) There is variation in the size of the Populus trichocarpa saplings (although in each group of trees the saplings are the same age). Measurements of $CH_4$ flux were taken for the canopy since we were testing for a relationship in general between flux and temperature and flux and isotopic fractionation. Higher flux can be due to larger tree size, but in that case, there is still a significant relationship between higher fluxes and temperature and higher fluxes and smaller fractionation. We suggest that this may be explained by the presence of different transport mechanisms, which could change with the maturity and size of the tree. Regardless, the relationship holds.

(ii) The volume variation of the Tedlar bag is addressed in lines 32-33. We calculated

the volume variation of the bag and found that it was not significant compared to the variation in $CH_4$ flux. That is, a +/- 10 L variation in bag volume would lead to a +/- difference of 0.15 ppm, which is well within +/- one standard deviation of the flux (0.42 ppm to 2.0 ppm).

(2) (i) The water temperature of the test chamber would have been in thermal equilibrium with the environment, as it was allowed to be in the greenhouse for 3-4 hours before the experiment. The temperature inside the tedlar bag may have varied; however, a difference in temperature between the green house and the interior of the bag during an experiment would have increased over time if they were initially at the same temperature. If this had a significant effect on flux, we should not have measured a generally linear relationship between $CH_4$ concentration in the bag and time.

(ii) We did combine our data sets and in spite of any physiological differences in the trees, the relationships between flux and temperature and fractionation held. This may demonstrate that these relationships are indeed independent of some physiological differences or genetic variation.

(3) (i) There is indeed a decrease of $CH_4$ flux with height of the main tree stem. Our estimation of total stem flux is an order of magnitude approximation to see if the canopy flux we measured with the Tedlar bag is comparable, since we were not able to measure flux from the entire stem. The order of magnitude is similar between the two, but we offer this only to suggest that our flux measurements could be explained by stem flux alone, not as proof.

(ii) Our calculations indicated that we should not be able to measure flux per leaf, as it would be below detection limits. Our trials with the leaf cuvette simply confirmed that no flux was in fact detectable, and were an auxiliary experiment.

(iii) The error shown for each data point in Fig. 3 is for the flux measurement. That is, each gas sample was divided into three parts run separately in the IRMS. The standard error is shown for each sample.

---

## Author Comment (AC2) · 6 May 2016

Thank you for your review of our manuscript. A reply to each point is made below.

It should be mentioned that our measurements of $CH_4$ flux from the canopies of Populus trichocarpa saplings are quantified to the end of finding any significant variation with temperature and isotopic fractionation. We do not intend the canopy measurements to be quantitatively comparable to measurements made under more precise conditions where the absolute value of the flux is sought.

As for vapor pressure deficits, these may have differed between the Tedlar bag and ambient. However, if $CH_4$ flux were mitigated in any way by a difference in vapor pressure deficit, it did not prevent the measurement of a significant relationship between flux, temperature and isotopic fractionation. This was the main goal of the investigation

and future work could include the study of other effects such as vapor pressure deficit on these variables.

(i) We did not measure the quantity of lenticels as we were not attempting to find any relationship between lenticel density and fractionation, only net flux, temperature and isotopic fractionation.

(ii) The saplings were grown under hydroponic conditions and were never in a soil environment. Water levels were regularly checked within the hydroponic system, and air was continuously bubbled through the growing containers.

(iii) We did not directly speculate on the implications of our results to the movement of gas through woody vegetation in landfills or in general, other than to say in the Discussion section that if $CH_4$ is present in the root zone our experiments demonstrate that it will be transported by the tree to the atmosphere. Our conclusions to this end are qualitative at this time.

(iv) We would be happy to provide the details of the modified Hoagland's solution, adding it to the manuscript.

(v) We did not measure the Tedlar bags for airtightness. There may have been a small amount of leaking from the bottom of the bag where it was secured to the tree. However, should any $CH_4$ have escaped from the bag, it did not impair our ability to establish the relationships between flux, temperature and fractionation.

(vi) The Tedlar bags did not include a mixing apparatus, although the stem chambers did contain a small fan. If mixing had been significantly poor in the Tedlar bag, we should not have measured a generally linear increase of $CH_4$ concentration in the bag over time.

———————————————————

---

## Referee Comment (RC3) · Anonymous Referee #3 · 8 May 2016

On the whole I don't have any issues with this papers approach, but I think that better referencing could have been used as there exist a number of studies in the literature that deal with the subjects discussed here which have importance. I am not trying to toot my own horn, or increase my citation index, but a lot of work has been done on this subject and I suspect the authors would want to know about it. Many of these articles are not available on instant PDF but they shouldn't be lost. For example, does the flux come from the leaves, or from the petioles? see Harden, H. and Chanton, J.P. 1994. Locus of methane release and mass dependent gas transport from wetland aquatic plants, Limnology and Oceanography, 39, 148-154. and also Whiting, G.J. and Chanton, J.P. 1996. Control of diurnal pattern of methane emission from aquatic macrophytes by gas transport mechanisms. Aquatic Botany, 54, 237-253.

fractionation can occur by a number of processes, mass flow and molecultar diffusion.

[Figure]

These vary with the amout of sunlight and possibly temperature. see Chanton, J.P., Whiting, G., Happell, J., and Gerard, G. 1993. Contrasting rates and diurnal patterns of methane emission from different types of vegetation. Aquatic Botany, 46, 111-128.

and Chanton, J.P., and G.J. Whiting. 1996. Methane Stable Isotopic distributions as indicators of gas transport mechanisms in emergent aquatic plants. Aquatic Botany, 54, 227-236.

The authors may be interested int Popp, T.J., J. P. Chanton, G.J. Whiting, N. Grant, 1999. The Methane Stable Isotope Distribution at a Carex Dominated Fen In North Central Alberta, Global Biogeochemical Cycles, 13, 1063-1077.

All of these paper discuss methane isotopic fractionation during gas transport. while not dealing with trees, the authors discuss many of the issues these authors are grappling with.

Finally the authors should look at this book. Trace Gas emissions by plants.

http://www.sciencedirect.com/science/book/9780126390100

On page 9, line 4, the authors state that gas transport through aerenchyma is usually diffusive... I would take issue with that. the authors suggest that diffusion is not the primary transport mechanism they have observed. They are not the first to think about this, and they might want to read the work of others who have thought about this too.

In conclusions they state that the methane is not associated with transpiration. That's right, I agree. We went to a lot of trouble to examine stomatal control of methane emission back in the 1990's. We couldn't find it either. Discuss.

---

## Author Comment (AC3) · 13 May 2016

I agree that many of these papers are excellent resources. Some have served as impetus for our research (Whiting and Chanton, 1996, and Chanton and Whting, 1996) though they were not directly cited in our manuscript.

The transport types that Whiting and Chanton studied were through-flow convection and diffusive transport, as is appropriate for aquatic macrophytes. A difference of humidity across a leaf boundary or thermal transpiration may induce through-flow convection. Armstrong and Armstrong (1991) found convective through-flow to be induced in the living leaf sheaths and stomata of Phragmites. This mechanism works via the connection between gas-filled spaces in the rhizomes and culms (stalks) of the plants, where gas flows in through younger culms and exits from older leaves and culms. Although we have not ruled out a similar mechanism in trees, the anatomy of Populus is different from that of these aquatic plants and may not support this kind of through-flow. In plants with convective through-flow, little isotopic fractionation was found (Chanton and Whiting, 1996). This also suggests our results are not indicators of this mechanism. Beckett et al. (1998) found that non-through-flow convection was likely to be of minimal importance in any submerged root system.

The second type of mechanism studied for aquatic plants and certainly in rice is diffusion. As pointed out in our manuscript, many researchers have cited diffusion as the transport mechanism associated with aerenchyma, although we did not come to this conclusion. Chanton and Whiting (1996) reported isotopic fractionation values between 4 and 16 permil for plants without convective through-flow. Our values are of similar range. However, Popp et al. (1999) and Chanton (2005) discuss the build-up of enriched methane as a factor that would decrease the overall isotopic fractionation, although this would be at steady state. Since our canopy methane concentration was increasing linearly with time, we cannot assume steady state in our system. Additionally, the work cited here involved measurements from plants in soils, where the influence of soil mechanisms needed to be taken into account. Our measurements should reflect tree transport alone. The temperature relationship with isotopic fractionation that we found did not match the temperature dependence of diffusion. Also, that isotopic fractionation decreased with increased flux and increased temperature suggests that more than one mechanism is at work, so that while diffusive transport may be present, we did not conclude that it is the dominant mechanism.

Our results are in agreement with studies demonstrating that stomata do not control methane emission. Again, however, tree physiology is different from that of aquatic plants. We believe methane is not leaving through the tree leaves in significant amounts and the aerenchyma pathway is dominant.